# TLR7: A Key Prognostic Biomarker and Immunotherapeutic Target in Lung Adenocarcinoma

**DOI:** 10.3390/biomedicines13010151

**Published:** 2025-01-09

**Authors:** Feiming Hu, Chenchen Hu, Yuanli He, Yuanjie Sun, Chenying Han, Xiyang Zhang, Lingying Yu, Daimei Shi, Yubo Sun, Junqi Zhang, Dongbo Jiang, Shuya Yang, Kun Yang

**Affiliations:** 1Department of Immunology, The Fourth Military Medical University, Xi’an 710032, China; 17778914060@163.com (F.H.); 18579121005@163.com (C.H.); yuanli5832@163.com (Y.H.); syjfly@163.com (Y.S.); hanchenying7777777@163.com (C.H.); sunyubo000103@163.com (Y.S.); zjq000211@163.com (J.Z.); superjames1991@foxmail.com (D.J.); 2Yan’an Key Laboratory of Microbial Drug Innovation and Transformation, School of Basic Medicine, Yan’an University, Yan’an 716000, China; 3Military Medical Innovation Center, The Fourth Military Medical University, Xi’an 710032, China; zhangxiyang199272@163.com; 4School of Basic Medicine, The Fourth Military Medical University, Xi’an 710032, China; 13467912072@163.com (L.Y.); 17330196403@163.com (D.S.)

**Keywords:** TLR7, lung adenocarcinoma, tumor microenvironment, tumor-infiltrating immune cells, prognosis, immunotherapy, candidate drugs

## Abstract

**Background:** The tumor microenvironment (TME) plays a crucial role in the progression of lung adenocarcinoma (LUAD). However, understanding its dynamic immune and stromal modulation remains a complex challenge. **Methods:** We utilized the ESTIMATE algorithm to evaluate the immune and stromal components of the LUAD TME from the TCGA database. Correlations between these components and clinical characteristics and patient prognosis were analyzed. Toll-like receptor 7 (TLR7) was identified as a key prognostic biomarker through PPI network and COX regression analysis. Validation of TLR7 expression was conducted using GEO data, qPCR, WB, and IHC. A prognostic model was developed using a nomogram, incorporating TLR7 expression. Enrichment analysis, the Tumor Immune Estimation Resource database, and single-sample gene set enrichment analysis were used to explore TLR7’s potential function. The response of the TLR7 subgroup to immunotherapy and drug sensitivity was observed. **Results:** We found significant associations between the immune and stromal components of LUAD TME and clinical features and prognosis. Specifically, TLR7 was identified as a prognostic biomarker, where lower expression in tumor tissues was linked to worse outcomes. This finding was further confirmed by comparing TLR7 expression in LUAD cells to normal bronchial epithelial cells, revealing lower expression in the tumor cells. Incorporating TLR7 into a nomogram prognostic model resulted in a good predictor of patient survival. Additionally, TLR7 was associated with immune function and positively correlated with various immune cells. Importantly, patients with high TLR7 expression were more likely to benefit from anti-PD-1 checkpoint blockade therapy. We also identified four treatment candidates for patients with high TLR7 expression. **Conclusion:** TLR7 is a powerful clinical feature that predicts patient prognosis, immunotherapeutic response, and drug candidates, providing additional insights for the treatment of LUAD.

## 1. Introduction

Lung cancer is the predominant contributor to mortality from cancer across the globe [1]. Lung cancer primarily categorizes into small cell lung cancer and non-small cell lung cancer (NSCLC). NSCLC encompasses various subtypes such as squamous cell carcinoma, adenocarcinoma, and large cell carcinoma, constituting approximately 80% of all lung cancers. Among these, adenocarcinoma is the most prevalent form of NSCLC [2]. Lung cancer has the second highest incidence rate of all cancers and the highest mortality rate, which has seriously affected people’s health [3]. Despite advances in treatment and the incorporation of new drugs and targeted therapies, the 5-year survival rate for LUAD patients is only 15% [4]. Patients are diagnosed earlier and provided with better treatment, which greatly improves survival rates [5]. The search for effective prognostic early biomarkers for individualized treatment has become imperative, and therefore, there is an urgent need for in-depth exploration of the mechanisms of lung cancer and development of new therapeutic approaches.

The tumor microenvironment (TME) consists of many different types of cells, including tumor cells, but also non-transformed cells such as fibroblasts, immune cells, neurons, and cells that form the lymphatic vessels and the vascular system [6,7] It plays an important role in the progression and immunotherapy of primary early-stage lung cancer [8]. For example, CAFs (cancer-associated fibroblasts) are able to limit the recruitment of immune effector cells such as CD8+T cells to tumor tissues by secreting different chemokines [9]. In addition, the proportion of immunosuppressive cells modified by CAFs, such as M2-type macrophages, is increased significantly. This change plays an important role in the process of tumor immunosuppression and contributes to the further progression and spread of tumor cells by reducing the activity of the immune system [10]. This suggests that changes in the tumor microenvironment may affect the nature of the tumor and may also influence the eventual development of the tumor: cancer-promoting or cancer-suppressing. The immune status of TME is an important factor influencing tumor progression, and the prognosis of tumors and survival time of patients can be significantly improved by immunotherapeutic strategies targeting TME [11]. In order to develop better therapeutic strategies, it is important to understand the mechanisms of tumor immune evasion and what leads to the altered state of the tumor microenvironment [12]. In the process of tumorigenesis and development, the continuous interaction between tumor cells and the tumor microenvironment plays a decisive role. This interaction not only promotes the proliferation and spread of tumor cells, but also influences their metastatic ability and response to therapeutic regimens. Therefore, targeting the tumor microenvironment for cancer therapy has attracted great interest in both the scientific and clinical fields, aiming to provide an important basis for the development of more precise and effective therapies through an in-depth understanding of this interaction mechanism [13]. Recent studies have also begun to use TME as a predictive biomarker to identify patients who may benefit from immunotherapy [14]. Early detection and treatment are critical for improving the prognosis of cancer patients, and adaptive immune responses within tumors have previously been shown to be strongest in the early stages of cancer. These findings emphasize the importance of finding immune biomarkers for early detection [15]. Therefore, we need to explore some immune-related genes that influence immune cell infiltration in TME at the early stage of tumorigenesis.

In our study, we analyzed and screened differentially expressed genes (DEGs) by comparing the immune and stromal components of TME. These DEGs provided us with important clues about the state of the tumor microenvironment and helped us further understand the mechanisms of tumor development. In this paper, we applied ESTIMATE, ssGSEA, and TIMER computational methods to calculate the composition of immune cells infiltrated in the tumor microenvironment and the ratio of immune and stromal fractions of LUAD samples from The Cancer Genome Atlas (TCGA) database, and identified a predictive biomarker, Toll-like receptor 7 (TLR7), which can be used to guide the prognosis and prediction of the response to immunologic and pharmacologic therapy. We validated the mRNA expression levels of TLR7 in human clinical specimens from TCGA and the GEO database. Using bioinformatics and prognostic models, combined with tumor-node-metastasis (TNM) staging of the tumors, the link between TLR7 expression and prognostic factors in LUAD was further elucidated. In conclusion, the discovery of TLR7 offers new hope for biomarkers and potential therapeutic targets for lung adenocarcinoma. This discovery not only provides a scientific basis for immunotherapy and drug therapy, but also opens up a new direction for targeted therapeutic strategies in TME.

## 2. Materials and Methods

### 2.1. Raw Data

RNA sequencing data and clinical profiles for 571 LUAD patients (with 58 normal and 513 tumor samples) were sourced from TCGA database (https://portal.gdc.cancer.gov/) (accessed on 30 December 2023). mRNA expression profiles and associated clinical details for lung adenocarcinoma were retrieved from the GEO database (https://www.ncbi.nlm.nih.gov/geo/) (accessed on 9 April 2024), encompassing datasets GSE10072, GSE32863, GSE68465, and GSE75037. Additionally, data on immune-related genes (IRGs) were accessed through the ImmPort platform (https://www.immport.org/home) (accessed on 7 March 2024).

### 2.2. Generation of ImmuneScore, StormalScore, and ESTIMATEScore

Employing the ESTIMATE algorithm within R version 4.3.2, we calculated the proportions of immune and stromal components within the tumor microenvironment (TME) for each sample. These proportions were quantified through three distinct scores: ImmuneScore, StromalScore, and ESTIMATEScore. Each score positively correlates with the respective ratios of immune cells, stromal cells, and their combined total within the TME, indicating that higher scores reflect a larger presence of the corresponding cellular fraction.

### 2.3. Survival Analysis

The survival analysis used the “survminer” (R packages (version 0.4.9)) and “survival” (R packages (version 3.5.7)) packages in R. Kaplan–Meier estimation was used for the curves, with a *p*-value threshold of less than 0.05.

### 2.4. Generation of DEGs Between High and Low Subgroups of ImmuneScores and StromalScores

A total of 513 tumor samples were grouped into high and low based on their ImmuneScore and StromalScore relative to the median. A DEG analysis using DESeq2 identified differentially expressed genes in high and low score groups. The criteria were log2 FC (fold change) greater than 1.0 and FDR below 0.05.

### 2.5. Heatmaps

The heatmap of DEGs was generated using the R package “pheatmap” in R language.

### 2.6. Analysis of Differences in Scores and Clinical Stages

Clinicopathological information for LUAD samples was sourced from the UCSC xena platform (https://xenabrowser.net/datapages/) (accessed on 30 December 2023) and the GEO database. The analysis was carried out in R, employing either the Wilcoxon rank sum test or the Kruskal–Wallis rank sum test as the statistical approach, depending on the number of clinical stages under comparison.

### 2.7. Functional Enrichment Analysis and Gene Set Enrichment Analysis

Gene Ontology (GO) and Kyoto Encyclopedia of Genes and Genomes (KEGG) enrichment analyses were performed using the “clusterProfiler” (R packages (version 4.10.0)) and “ggplot2” R packages (version 3.5.1). The GO analysis encompassed molecular function (MF), biological process (BP), and cellular component (CC) categories. Enrichment was considered significant for terms with both *p*- and q-values below 0.05. The complete transcriptomes of all tumor samples were analyzed to identify potential mechanisms influencing prognostic genes using gene set enrichment analysis (GSEA), which also helped in determining the differentially regulated pathways. A false discovery rate (FDR) of less than 0.05 was set as the threshold for statistical significance.

### 2.8. Weighted Gene Co-Expression Network Analysis

Gene co-expression networks were established utilizing the WGCNA-R package to investigate the relationship between gene networks and scores such as ImmuneScore, StromalScore, and tumor purity. This method identified the top 546 genes for further study. The optimal soft threshold β was selected through the “pickSoftThreshold” function, which assesses the neighborhood matrix to determine key modules. Pearson’s correlation coefficients were computed for each module in relation to the ImmuneScore, StromalScore, and tumor purity score, with the most significantly correlated modules being chosen for deeper analysis.

### 2.9. Protein–Protein Interaction (PPI) Network and COX Regression Analysis

Protein–protein interaction (PPI) networks were analyzed on the STRING platform to determine gene interactions (https://cn.string-db.org/) (accessed on 12 May 2024). Key genes were pinpointed in Cytoscape version 3.9.0, setting a confidence threshold of 0.70. Univariate Cox regression analysis was conducted with the survival package in R to identify prognostic genes, considering *p*-values less than 0.05 as significant.

### 2.10. Cell Culture

The BEAS-2B bronchial epithelial cells were cultured in DMEM medium enriched with 10% FBS. Meanwhile, the human LUAD cell lines, A549, H1299, and PC9, were cultivated in RPMI-1640 medium with the same 10% FBS (fetal bovine serum) supplementation. Regular authentication and mycoplasma contamination checks were conducted to ensure the integrity of all cell cultures.

### 2.11. Quantitative Real-Time Reverse Transcription Quantitative PCR

Total RNA was extracted from the prepared cells using an RNA extraction kit from TSINKGE in Beijing, China. This RNA was then converted to cDNA by HiScript ll Q RT SuperMix for qPCR (+gDNA wiper) (Vazyme Biotech, Nanjing, China). For the subsequent real-time quantitative PCR (qPCR), the SYBR Green PCR Master Mix from Vazyme Biotech in Nanjing, China, was utilized. The primer sequences employed in this process are detailed in Appendix A.

### 2.12. Western Blot and Immunoprecipitation Assays

Protein expression was measured by Western blot as previously described [16]. Following blocking, the polyvinylidene difluoride membranes were incubated overnight at 4 °C with the various antibodies listed in Appendix A.

### 2.13. Immunohistochemistry (IHC) Analysis

TLR7 expression was detected by immunohistochemistry (Abways, Shanghai, China, CY6812, 1/100) in human lung adenocarcinoma and its adjacent tissues as previously described [17].

### 2.14. Nomogram Construction

Multivariate Cox regression analysis was used to develop prognostic models and risk score equations, and the area under the curve of LUAD patient characteristics was used to evaluate the sensitivity and specificity of the models. The models’ sensitivity and specificity were assessed by calculating the area under the curve (AUC) for LUAD patient characteristics. The discrimination power of the survival models was gauged using the Consistency Index (C-index), with higher values signifying greater discrimination ability. Nomograms were created based on the identified prognostic factors to forecast 1-, 3-, and 5-year survival probabilities.

### 2.15. Immune Cell Infiltration Analysis

We utilized the single-sample gene set enrichment analysis (ssGSEA) approach within the GSVA R package (version 1.50.1) and the Tumor Immune Estimation Resource (TIMER) (http://timer.cistrome.org/) (accessed on 23 May 2024) to assess immune cell infiltration in lung adenocarcinoma. This research focused on the impact of TLR7 expression on the infiltration of immune cells, leveraging gene expression data. To explore the relationship between TLR7 expression levels and the presence of tumor-infiltrating immune cells, statistical significance was determined using both the Wilcoxon rank sum test and the Spearman rank correlation test for *p*-value calculation.

### 2.16. Immunotherapy and Prediction of Drug Sensitivity

The TIDE and “oncoPredict” R packages (version 1.2) were employed for forecasting immunotherapy responses and drug sensitivities, respectively. TIDE data can be accessed at http://tide.dfci.harvard.edu/login/ (accessed on 18 June 2024), while IPS scores were sourced from the TCIA database, available at https://tcia.at/home (accessed on 18 June 2024).

## 3. Results

### 3.1. Analysis of the Correlation Between Score and Clinical Stage and Survival Rate of LUAD Patients

We initially collected 513 tumor samples from the TCGA database. The samples were divided into high and low groups based on the median value of each score, and Kaplan–Meyer survival analyses were performed on the ImmuneScore, StromalScore, and ESTIMATEScore. The ESTIMATEScore is the sum of the immune and stromal scores, and reflects the combined proportions of these two components in the TME. The results showed that lung adenocarcinoma patients with higher ImmuneScore (Appendix A, *p* = 0.012), StromalScore (Appendix A, *p* = 0.016), and ESTIMATEScore (Appendix A, *p* = 0.01) had significantly longer survival times. In addition, the relationship between these three scores and clinical characteristics was investigated using clinical data extracted from the UCSC Xena database. To determine the relationship between the proportion of immune and stromal components and clinicopathological staging, we defined T1 and Stage I as early stage and T2–4 and Stages II–IV as intermediate to late stage, and then analyzed the corresponding clinical information of LUAD cases in the TCGA database. ImmuneScore showed a greater immunity score with T1 than T2–4 in TNM staging (Appendix A, *p* < 0.001), a greater immunity score with M0 than M1 in TNM staging (Appendix A, *p* = 0.03), and a greater immunity score with Stage I than Stage II–IV in TNM staging (Appendix A, *p* = 0.019), which suggests that ImmuneScore was negatively correlated with T-staging, distant metastases, and stage grading; StromalScore was greater for M0 than M1 in TNM staging only, which may be linked to the M classification of clinical information (Appendix A, *p* = 0.005); and ESTIMATEScore declined significantly with progression of T-staging, distant metastases, and stage grading (Appendix A, *p* = 0.005; Appendix A, *p* = 0.008; Appendix A, *p* = 0.015). The findings indicate a correlation between LUAD stage and the proportion of immune and stromal components within the TME (Figure 1).

### 3.2. Screening for Immune-Related Differential Genes Expression

In order to ascertain the precise alterations in gene expression profiles with regard to immune and stromal elements within the TME, an evaluation of mRNA sequencing data was conducted on 513 patients with LUAD from the TCGA, with a comparison of samples having high and low scores. Compared to the median, a total of 1284 DEGs (high and low scoring samples) were obtained from ImmuneScore, including volcano and heatmaps for 728 upregulated genes and 556 downregulated genes (Figure 2A,B). Similarly, 1093 DEGs were obtained from StromalScore, including volcano and heatmaps for 593 upregulated genes and 500 downregulated genes (|log2FC| > 1, *p* < 0.05) (Figure 2C,D). Venn diagrams showing crossover analyses revealed that 257 high-scoring upregulated genes were co-expressed in ImmuneScore and StromalScore (Figure 2E), and a total of 327 genes exhibited downregulation (Figure 2F).

### 3.3. WGCNA Identification of Gene Modules Related to StromalScore, ImmuneScore, ESTIMATEScore, and TumorPurity Score

Firstly, the samples were selected for hierarchical cluster analysis, and after excluding the abnormal data, 513 samples were finally used to construct co-expression networks through the R package “WGCNA”, which revealed the gene modules associated with StromalScore, ImmuneScore, ESTIMATEScore, and TumorPurity score. In this study, β = 10 was the best choice of soft threshold for constructing the scale-free network (Figure 2G,H). After adjusting the WGCNA parameters, the phylogenetic tree was used to mine the co-expression modules (Figure 2I), and finally eight modules were obtained and the correlation between the feature vectors of these eight modules and the StromalScore, ImmuneScore, ESTIMATEScore, and TumorPurity score were calculated (Figure 2J). From this, it can be seen that the module correlated most closely with the three ESTIMATE scores, with an average correlation coefficient > 0.8, and had the strongest negative correlation with the TumorPurity scores, obtaining 546 correlated genes. We took the intersection of genes upregulated in ImmuneScore and StromalScore, and genes related to the MEturquoise module, and a total of 117 DEGs were identified in the Venn diagram (Figure 2K). These DEGs (117 genes in total) may be determinants of TME status. We then performed GO functional enrichment analysis on these 117 genes. The results show the top 20 biological processes (BP) (Figure 2L). The function of differentially expressed genes correlates with immune activity, indicating that immune factors play a key role in the lung adenocarcinoma tumor microenvironment; KEGG analysis yielded pathways, most of which also showed enrichment of immune-related pathways (Figure 2M). In summary, the combined functions exhibited by these DEGs mapped qualities that were closely associated with immune activities, which further revealed that immune factors play an important role in the tumor microenvironment of LUAD. Additionally, from the previous analysis, the infiltration, metastasis, and stage of LUAD were associated with the more likely proportion of immune components in the tumor microenvironment, so we took the intersection of these co-expressed genes with IRGs (immunologically relevant list of genes) and we found a total of 38 DEGs from the Venn diagrams (Figure 2N).

### 3.4. PPI Network Analysis and Univariate COX Regression

To delve deeper into the underlying mechanisms, we established PPI networks utilizing the STRING with the aid of Cytoscape software (version 3.9.0) interactions between genes (Figure 3A), and the bars represent the top 20 genes ranked by number of nodes (Figure 3B). A univariate Cox regression was used to analyze the survival of LUAD patients and identify prognostic genes, and we identified 12 significant genes (Figure 3C). Next, we analyzed the top 20 genes and 12 factors for correlation., with a total of five molecules co-expressed, namely, TLR7, PTPRC, IL7R, ICOS, and IL10RA (Figure 3D).

### 3.5. Association of TLR7 Expression with Clinical Characteristics and Survival in LUAD Patients

The objective of this study is to elucidate the relationship between toll-like receptor 7 and the clinical characteristics of patients with lung adenocarcinoma. The result showed that TLR7 was significantly downregulated in LUAD tissues compared to normal lung tissues (Figure 3E). The expression of TLR7 in paired LUAD tissues was found to be significantly lower than in normal tissues (Figure 3F). We obtained the same results from three more GEO datasets (GSE10072, GSE32863, GSE75037) (Appendix A respectively). In particular, the expression of TLR7 was found to be positively correlated with tumor size. Additionally, TLR7 expression in Stage I exhibited higher TLR7 expression than Stages II–IV (Figure 3G), and Stage T1 was significantly higher compared to stages T2–4 (Figure 3H), which is consistent with our findings in GSE75037 (Appendix A). In addition, we also verified at the cellular and tissue levels by qPCR, WB, and IHC that TLR7 expression was higher in normal bronchial epithelial cells and adjacent normal tissues than in LUAD cell lines and lung adenocarcinoma tissues (Figure 3I–M). We analyzed the link between TLR7 and clinical characteristics; patients with LUAD and high TLR7 expression had longer survival times (Figure 3N). We then analyzed the relationship between TLR7 expression and survival from the GSE68465 dataset and obtained the same results (Appendix A). TLR7 high expression in the TME is linked to better LUAD patient outcomes.

### 3.6. Establishment of TLR7-Related Prognostic Model and Immunotherapy Sensitivity Analysis

In this study, we utilized the R package rms to integrate data on seven gender-related characteristics, including stage grading, T stage, lymph node metastasis, distant metastasis, smoking status, and TLR7 gene expression level, as well as survival time and survival status. We excluded clinical samples with unknown or unclear information and constructed a nomogram using the Cox method. As shown in Figure 4A, we incorporated seven indicators to build the prognostic model, and based on the patient’s own situation, we could find the corresponding scores and add these scores to obtain the total score to assess the corresponding one, three, and five-year survival rate of the patient. Moreover, the prognostic significance of these characteristics was evaluated in a total of 340 samples. The overall C-index of the model was 0.697754679442933, 95%CI (0.654281245010108, 0.741228113875758), *p* = 4.85151974751457 × 10^−19^. The nomogram showed how each variable influenced survival through the length of its line. The main factor in cancer survival was TLR7, with T and N stage next, and gender, stage, and smoking status were the least important (Figure 4A). At the same time, we plotted calibration curves with clinical factors to verify the accuracy of the prediction model (Figure 4B). The satisfactory agreement observed between the predicted and actual values indicates that our model holds promise for prognostic prediction in LUAD patients. Subsequently, we conducted LUAD ROC analysis using the R software package pROC (version 1.17.0.1) to calculate the AUC. Specifically, we collected data on patient’s gender, stage grade, T, N, M, smoking score, and TLR7 gene expression for ROC analyses at 1, 3, and 5 time points utilizing pROC’s ROC function. The AUC values and CI were determined using the ci function of the pROC package (Figure 4C). Patients were categorized into two cohorts according to their median risk scores, and the survival analysis package’s “survfit” function in R was used to further scrutinize the prognostic disparities between these cohorts. The log-rank test was applied to evaluate the statistical significance of the survival differences among the groups, revealing that the high-risk group survived less than the low-risk group (Figure 4D). To assess how various risk subgroups might respond to immunotherapy, we employed TIDE to gauge the potential effectiveness of immunotherapy among patients with varying risk profiles. Specifically, a reduced TIDE score suggested a decreased probability of immune evasion, which implies a higher chance of benefiting from immunotherapy. The lower TIDE scores in the low-risk group compared to the high-risk group indicated that the former might be more responsive to immunotherapy. In our findings, the low-risk group exhibited lower TIDE scores (Figure 4E). This suggests low-risk patients may respond better to immunotherapy, so they might be suitable for immunotherapy.

### 3.7. TLR7 Expression Is Associated with the LUAD Tumor Microenvironment

As previously mentioned, TLR7 is an immune-related gene, and given that TLR7 levels are negatively correlated with LUAD patients and T-staging and stage grading, they are also linked with survival in LUAD patients. Subsequently, we divided into TLR7 high expression group and TLR7 low expression group according to the median expression of TLR7 for differential gene analysis. (Figure 5A,B). In order to explore the function and pathway changes in these differential genes, we analyzed these differential genes by GO and KEGG, and found that almost all of them were related to immunity and metabolism, both in terms of function and pathway (Figure 5C,D). Therefore, we subsequently performed GSEA to further investigate the disparities in enrichment pathways between the high and low TLR7 expression groups. Notably, it is noteworthy that the group with high expression of TLR7 exhibited significant enrichment of pathways related to the immune system (Figure 5E; Appendix A). Conversely, in the low TLR7 expression group, metabolic pathways were predominantly enriched (Figure 5F; Appendix A). Importantly, the downregulation of TLR7 expression was associated with a shift from an immunologically dominant tumor microenvironment to a metabolically dominant state. These results suggest that TLR7 may serve as a promising TME marker influencing the tumor microenvironment.

### 3.8. Relationship Between TLR7 and Immune Cell Infiltration in LUAD

TILs (Tumor Infiltrating Lymphocytes) in TCGA lung adenocarcinoma samples were determined using the TIMER algorithm, and we analyzed how changes in TIL components correlate with TLR7 expression. Our findings indicated that variations in the gene copy number of TLR7 significantly influenced the levels of infiltration by B cells, CD8+ T cells, CD4+ T cells, macrophages, neutrophils, and dendritic cells (Figure 6A). TLR7 inversely correlated with tumor purity, and positively with B cells, CD8+ T cells, CD4+ T cells, macrophages, neutrophils, and dendritic cells (Figure 6B). The survival analysis revealed a link between lung adenocarcinoma survival and B cells, DC cells, and TLR7 expression (Figure 6C). The RNA sequencing data from these LUAD samples were analyzed using the ssGSEA method. We identified and selected 25 immune cell types with an abundance greater than 0 to evaluate the level of immune cell infiltration (Figure 6D). Figure 6E illustrates our discovery that the majority of immune cell populations were elevated in the group with high TLR7 expression compared to the low-TLR7-expression group, as determined by ssGSEA analysis. This also included B cells, CD8+ T cells, CD4+ T cells, macrophages, neutrophils, and dendritic cells. Despite the presence of immunosuppressive cells like MDSCs in the high-TLR7-expression group, the predominant immune cells with tumoricidal potential were more abundant, suggesting that TLR7 acts as an oncogene. Subsequent correlation analysis based on ssGSEA outcomes demonstrated a substantial link between TLR7 expression levels and 27 different tumor-infiltrating immune cell types (Figure 6F). In addition, macrophages, mast cells, memory B cells, monocytes, and various immune cell subtypes such as natural killer T cells (NKT) and plasmacytoid dendritic cells were also found to be positively correlated with TLR7 expression (Appendix A). The differential and correlation analyses collectively demonstrated that a total of twenty-four distinct types of tumor-infiltrating immune cells exhibited significant correlations with the expression levels of TLR7 (Appendix A). These findings further support the notion that alterations in TLR7 levels influence the immune activity within the tumor microenvironment (TME). We validated these results using two additional GEO datasets namely GSE32863 and GSE75037, except for specific subsets including CD56bright Natural Killer Cells, CD56dim Natural Killer Cells, Central Memory CD4 T Cells, Effector Memory CD4 T Cells, Immature Dendritic Cells, and Mast Cells. Importantly, our findings from these datasets were consistent with those obtained from analyzing data from TCGA (Appendix A).

### 3.9. Analysis of Drug Sensitivity and Immunotherapy for High and Low TLR7 Expression

To investigate the role of TLR7 in drug therapy, we utilized the R software package “oncoppredict” to predict the drug sensitivity of commonly used drugs in LUAD. A total of 513 cases of LUAD were obtained from the TCGA database, and an additional 58 cases were acquired from the GSE75037 dataset in the GEO database. Based on patients’ median TLR7 expression levels, they were categorized into high- and low-TLR7-expression groups. We identified that 96 drugs in TCGA and 4 drugs in GSE75037 exhibited increased sensitivity towards patients with high TLR7 expression, leading to enhanced drug efficacy (Figure 7A,B). Subsequently, a Venn diagram was drawn to identify common sensitive drugs between TCGA and GSE75037 datasets, revealing four shared drugs: Doramapimod, PF-4708671, AZD6482, and MBS-754807 (Figure 7C). These findings suggest that these four drugs demonstrate superior therapeutic effects for LUAD patients with high TLR7 expression levels, thus holding significant implications for guiding clinical medication decisions and improving treatment accuracy. Immune checkpoints were obtained from the TCIA database by Immunophenoscore (IPS), to determine the immunogenicity of LUAD and predict its response to immune checkpoint inhibitor therapy. A higher IPS indicates a more favorable effect of immune checkpoint blockade and increased patient sensitivity to immunotherapy. Notably, ips_ctla4_neg_pd1_pos (low level of CTLA-4 expression (neg) and high level of PD-1 expression (pos)) and ips_ctla4_pos_pd1_pos (high level of CTLA-4 expression (neg) and high level of PD-1 expression (pos)) with high TLR7 expression exhibited significantly higher levels compared to those with low TLR7 expression, suggesting that patients with high TLR7 expression may benefit more from anti-PD-1 therapy than CTLA-4 inhibition (Figure 7D). These findings hold valuable implications for personalized treatment strategies targeting LUAD patients with high TLR7 expression in terms of immunization and drug interventions.

## 4. Discussion

By 2022, there will be 482 million cancer cases and 321 million deaths in China. Lung cancer will account for most cancer deaths [18]. With the advancement of medicine, more and more therapeutic methods have been utilized in the clinic, including advances in surgery, radiotherapy, and molecular therapy techniques, which have led to a significant improvement in the clinical prognosis of LUAD patients [19,20,21,22]. However, the five-year overall survival rate for patients with lung adenocarcinoma (LUAD) is approximately 18% [3].

Before the advent of immunotherapy, the standard of care was a platinum double agent combined with carboplatin, or cisplatin combined with gemcitabine, vincristine, or paclitaxel. Several studies have concluded that these dual agents have comparable efficacy in patients with non-small cell lung cancer, but these drugs have significant side-effects [23]. Certain immunotherapies are so effective compared to standard chemotherapy that they have the potential to become the new standard of care for patients with advanced non-small cell lung cancer [24]. With the discovery of immunotherapy, the treatment paradigm for patients with advanced lung cancer has fundamentally changed the treatment of lung cancer and is still evolving [25]. Immunotherapy offers a revolutionary approach to fighting lung cancer. Notably, immunotherapy has fewer side-effects than chemotherapy, which not only prolongs patients’ survival but also improves their quality of life [26]. Chemotherapy directly destroys tumor cells and enhances the effectiveness of the immune system by boosting the number of T-cells relative to cancer cells; in addition, chemotherapy reduces the immunosuppressive substances released by the tumor, thereby reducing the suppression of the immune response; finally, chemotherapy helps to release tumor antigens, which activate the immune system and work synergistically with immunotherapy in the fight against tumors [27]. Chemotherapy can also stimulate PD-L1 expression in non-small cell lung cancer [28]. Chemotherapy in combination with immunotherapy can improve the therapeutic effect on lung adenocarcinoma. However, the tumor microenvironment plays a critical role in cancer development and progression and is involved in resistance to chemotherapy and immunotherapy [29]. Therefore, in order to more accurately predict the survival of LUAD patients and their response to immunotherapy and drug treatment, there is an urgent need to discover new biomarkers to improve the state of the tumor microenvironment, which in turn improves this resistance phenomenon and improves the therapeutic efficacy. The therapeutic goal for patients with early-stage lung cancer is a cure, and understanding the characteristics of the tumor microenvironment in early-stage invasive lung adenocarcinoma may be important for early detection and intervention, which may facilitate the development of some new therapeutic approaches [30]. In this study, we conducted an in-depth analysis of TLR7, an immune-related gene in the tumor microenvironment (TME), using extensive bioinformatics methods, and found that this gene was effective in predicting the treatment efficacy and prognosis of patients with LUAD, and had a predictive value for the response to immunotherapeutic and chemotherapeutic agents.

TLR7 is a transmembrane glycoprotein expressed on the inner membrane of certain immune cells [31,32,33]. Upon ligand binding, TLR7 undergoes a conformational change that leads to the expression of nuclear factor kB-mediated inflammatory cytokines, such as interleukin 6 and type 1 interferon [34,35]. TLR7 agonists are capable of inducing certain cytokine responses that promote changes in the tumor microenvironment and enhance the cytotoxicity of systemic antitumor CD8+ T cells [36]. Although MDSCs suppress the immune system, TLR7 agonists can differentiate them into macrophages and dendritic cells, improving cancer immunotherapy [37]. TLR7 agonists can be effective in potentiating antitumor effects, and there is growing evidence that these agonists can directly stimulate antitumor responses in a variety of cancers when used in combination with standard therapies. For example, preclinical studies have been conducted in various tumor models, including lymphoma, colorectal cancer, fibrosarcoma, and pancreatic cancer [38,39]. The antitumor effect of TLR7 agonists alone or in combination with immunotherapy was mainly dependent on the increase in the number of infiltrating dendritic cells, M1 macrophages, CD4+ T cells, and CD8+ T cells in the tumor microenvironment, remodeling the tumor microenvironment, and significantly inhibiting tumor growth in colorectal cancer and melanoma-bearing mice, especially in tumor types where immune checkpoint blockers alone were ineffective, and these compounds enhance the antitumor efficacy of PD-1/PD-L1 inhibitors. TLR7 agonists may augment immunotherapy by augmenting antitumor immune responses [40]. TLR7 plays a pivotal role in regulating immune cell ratios within the tumor microenvironment to exert antitumor functions.

TLR7 expression is reduced in lung adenocarcinoma. This influences progression and is an independent LUAD prognosis factor. Subsequently, we validated the expression of TLR7 in lung adenocarcinoma and its effect on survival in the GEO, which further refined our findings. GSEA analysis revealed that the TLR7 high-expression group was mainly associated with immune functions and pathways, whereas the TLR7 low-expression group was mainly associated with metabolic pathways, and it is possible that it is this shift that affects the immune infiltration and distribution status in the TME, which in turn affects tumor progression. This finding offers new insights on how to treat LUAD as TME status transitions. The model was analyzed for sensitivity to immunotherapy using the TIDE in high- and low-risk groups. The low-risk group was more sensitive to immunotherapy, with a lower TIDE score. TLR7 influences LUAD patients’ prognosis. We analyzed TLR7 functions using GO, KEGG, and GSEA. TLR7 expression in lung adenocarcinomas correlates with immune infiltration levels, including B cells, CD4+ T cells, CD8+ T cells, dendritic cells, and macrophages. This suggests a key role for TLR7 in the TME. The above results suggest that TLR7 may inhibit tumorigenesis and progression by increasing the infiltration rate and distribution of TIL. The results of drug sensitivity and response to immunotherapy calculated with “oncoPredict” and IPS showed that individuals with elevated TLR7 were more responsive to immunotherapy, and individuals with elevated TLR7 levels were more likely to benefit from anti-PD-1 checkpoint blockade therapy. In addition, we identified four promising therapeutic agents (doramapimod, PF-4708671, AZD6482, and MBS-754807) tailored to patients with high TLR7 expression. In conclusion, TLR7 is a reliable clinical indicator for predicting patient prognosis, immunotherapy response, and potential drug targets, thus providing new perspectives on therapeutic approaches for lung adenocarcinoma.

## 5. Conclusions

TLR7, a promising immune-related gene, predicts prognosis, response to immunotherapy, and agents in LUAD. It categorizes patients into subgroups with significant survival differences in TCGA and GSE68465, informing clinical practice. Notably, B cells, CD4+ T cells, CD8+ T cells, dendritic cells, and macrophage cells were positively correlated with TLR7 expression. In addition, patients with elevated TLR7 levels derived more benefit from anti-PD-1 than CTLA-4 check-point blockade therapy. Meanwhile, we optimized four potential therapeutic agents (Doramapimod, PF-4708671, AZD6482, and BMS-754807) to treat patients with high TLR7 expression. This study helps us understand how tumor stage and prognosis affect the tumor immune microenvironment. This could lead to new ways of treating patients with LUAD.

## Figures and Tables

**Figure 1 biomedicines-13-00151-f001:**
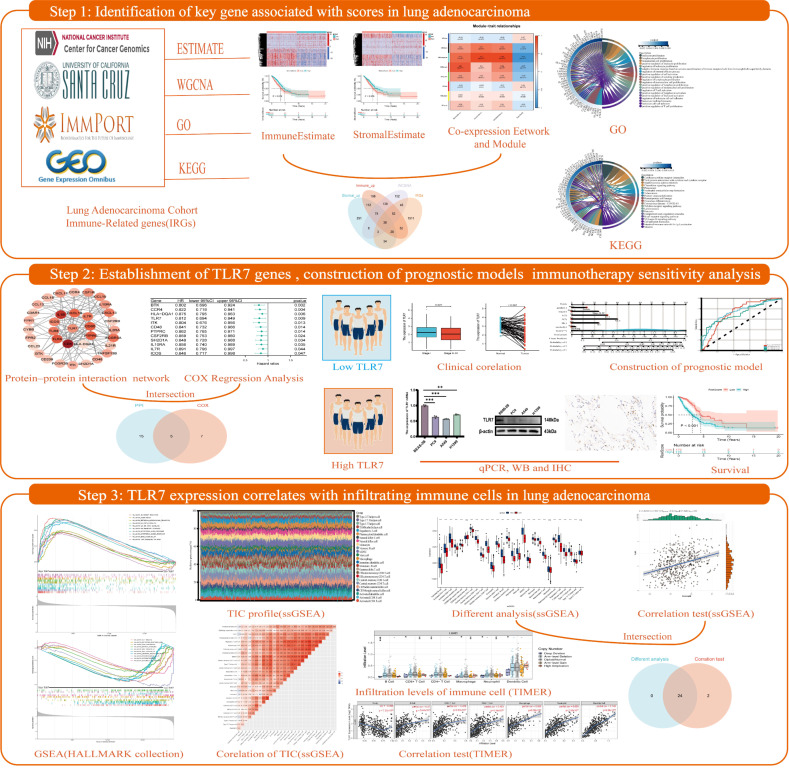
Flow chart of the framework for this study. Step 1: An in-depth analysis of lung adenocarcinoma samples from the TCGA database using the ESTIMATE algorithm revealed significant correlations between scores on immune and stromal components of the tumor microenvironment and clinical features and patient prognosis, and identified key genes associated with these scores. Step 2: TLR7, identified by PPI and Cox analysis, is a key biomarker for LUAD prognosis, and lower tumor expression is associated with poorer prognosis, as validated by GEO. qPCR, WB, and IHC confirmed that TLR7 expression is lower in LUAD compared to normal cells. A nomogram-based prognostic model incorporating TLR7 can effectively predict patient survival. Step 3: By enrichment analysis, TLR7 was found to be closely associated with immune function. Further immune infiltration analysis showed that TLR7 was positively correlated with the infiltration levels of multiple immune cells. These findings may explain the ability of the tumor microenvironment (TME) to maintain an immunodominant state. Step 4: The response of TLR7 subgroups to immunotherapy was analyzed and patients with high TLR7 were found to be more likely to benefit from anti-PD-1 checkpoint blockade therapy. Four therapeutic candidates for patients with high TLR7 were identified.

**Figure 2 biomedicines-13-00151-f002:**
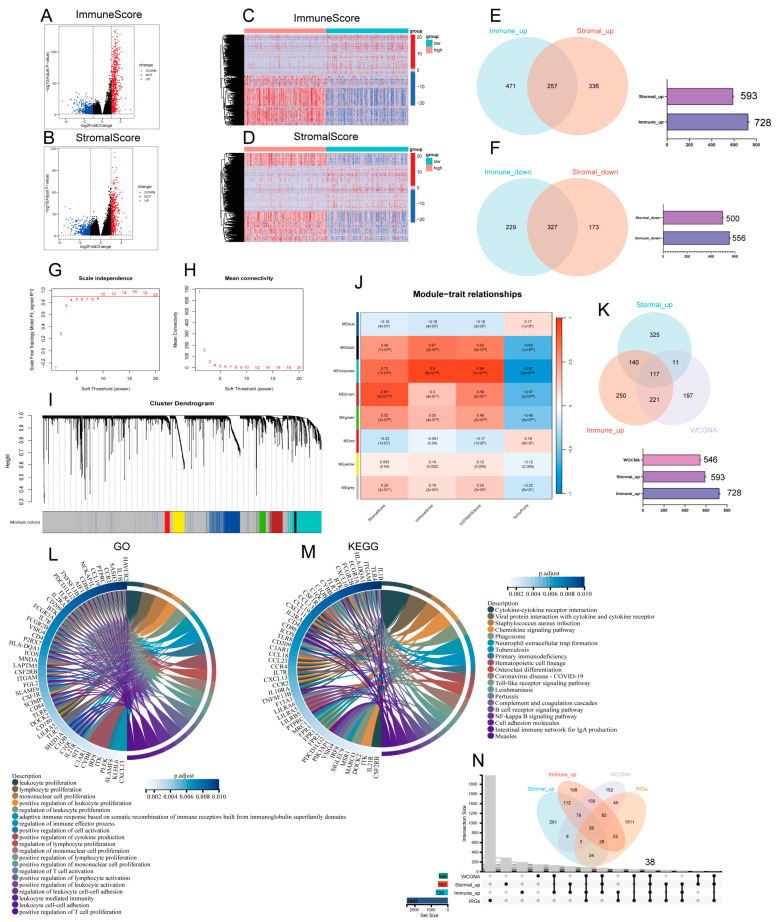
Screening for immune-related differential genes. (**A**,**B**) Graphical representation of differentially expressed genes (DEGs) as a volcano plot, contrasting the high and low groups in terms of ImmuneScore and StromalScore. (|log2FoldChange| >1 and adjusted *p*-value < 0.05). (**C**,**D**) Graphical representation of DEGs as a heat plot, contrasting the high and low groups in terms of ImmuneScore and StromalScore. (**E**,**F**) Venn diagrams illustrating the overlap of DEGs that are either up- or downregulated in both ImmuneScore and StromalScore. (**G**,**H**) TCGA-LUAD cohort scale independence and average connectivity. (**I**) The cluster dendrogram clusters similar gene expressions into modules, where each color represents a gene. (**J**) Pearson correlation analysis of the merged modules with ImmuneScore, StromalScore, ESTIMATEScore, and TumorPurity. (**K**) The Venn diagram illustrates the DEGs common to upregulation and WGCNA in ImmuneScore and StromalScore. (**L**) Circular plots depicting the GO enrichment analysis for 117 DEGs, highlighting terms significantly enriched at a *p*-adjusted threshold of less than 0.05, focusing on the top 20 GO-BP terms. (**M**) Circular plot for KEGG enrichment analysis of 117 DEGs, featuring terms significantly enriched at a *p*-adjusted threshold of less than 0.05, with an emphasis on the top 20 KEGG terms. (**N**) UpsetR and Venn plots of previously filtered immune and stromal shared upregulated DEGs and related genes screened by WGCNA and IRGs.

**Figure 3 biomedicines-13-00151-f003:**
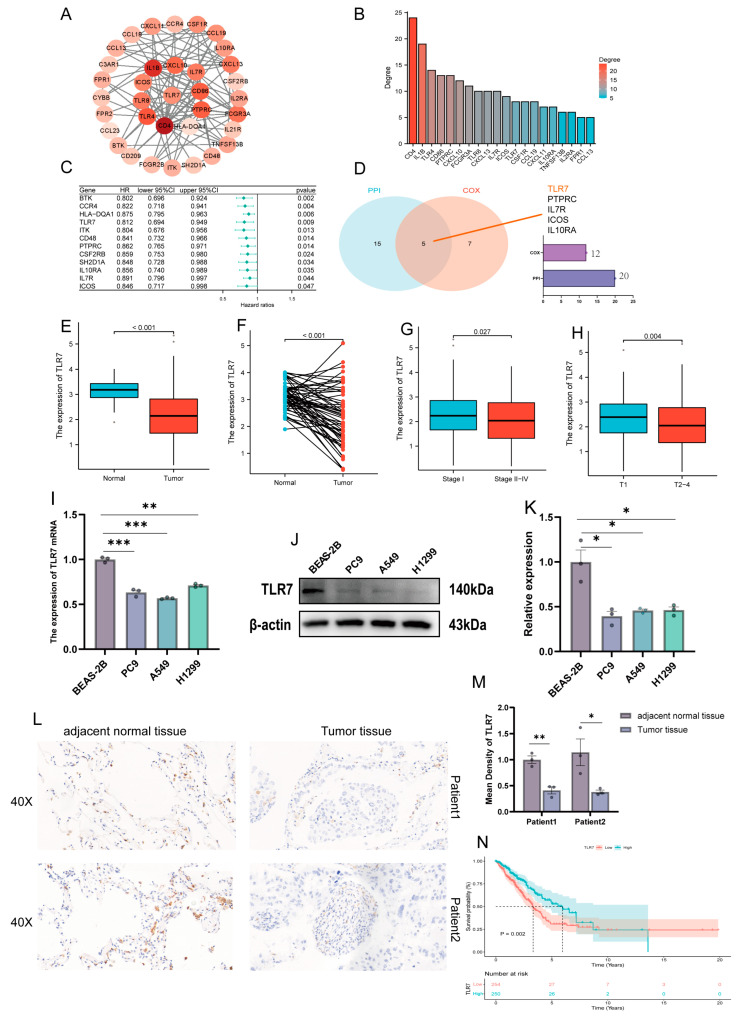
PPI and univariate Cox analysis of DEGs, and association of TLR7 expression with clinical characteristics and survival. (**A**) A network with nodes above 0.7 was constructed. (**B**) The 20 most significant genes. (**C**) A univariate Cox regression on 38 DEGs identified 12 genes with *p* < 0.05. (**D**) The Venn diagram shows the factors common to the 20 top PPI nodes and the univariate Cox model. (**E**) TCGA data were analyzed in all normal and tumor samples for TLR7 expression. (**F**) TCGA data were analyzed in paired normal and tumor samples from the same patient for TLR7 expression. (**G**) Association of TLR7 expression with stage classification in early and mid-late clinical stages. (**H**) Association of TLR7 expression with early and mid-late clinical T stage. (**I**–**K**) TLR7 expression at mRNA and protein levels in bronchial epithelial cells and lung adenocarcinoma cell lines. (**L**,**M**) TLR7 expression in paraneoplastic and tumor tissues. (**N**) Survival analysis of LUAD patients with different TLR7 expression, marking patients as high or low expression based on comparison with the median expression level. *p* = 0.002 by log-rank test. ***: *p* < 0.001, **: *p* < 0.01, *: *p* < 0.05.

**Figure 4 biomedicines-13-00151-f004:**
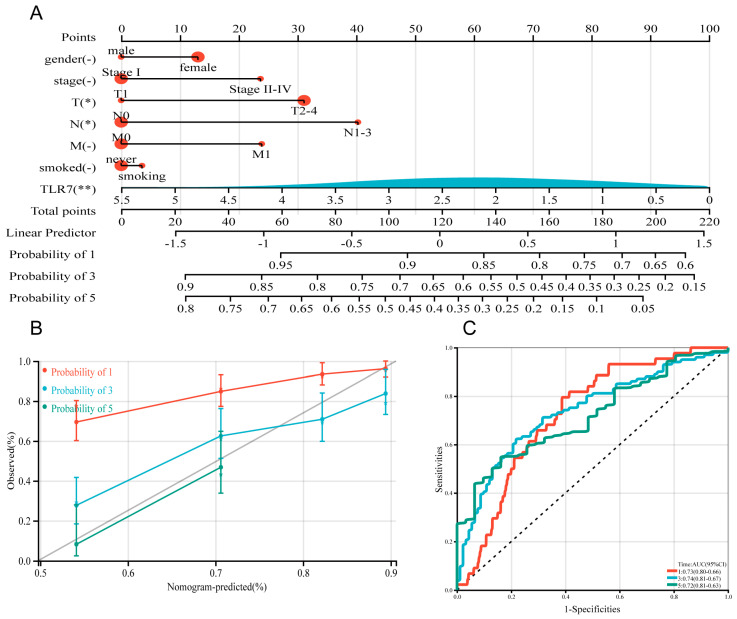
Construction and calibration of a nomogram. (**A**) A prognostic nomogram was developed by integrating TLR7 with multiple clinical variables to predict survival in LUAD patients. (**B**) A calibration plot assessed the accuracy of the prognostic model for estimating survival probabilities. (**C**) The area under the ROC curve, AUC, was used to predict the overall survival of LUAD patients at 1, 3, and 5 years. (**D**) Patients were categorized into high-risk and low-risk groups based on the median RiskScore expression level, and the log-rank test was used for survival analysis, *p* < 0.001. (**E**) TIDE scores in different risk subgroups, by *t*-test, *p* < 0.001. **: *p* < 0.01, *: *p* < 0.05, -: not significant.

**Figure 5 biomedicines-13-00151-f005:**
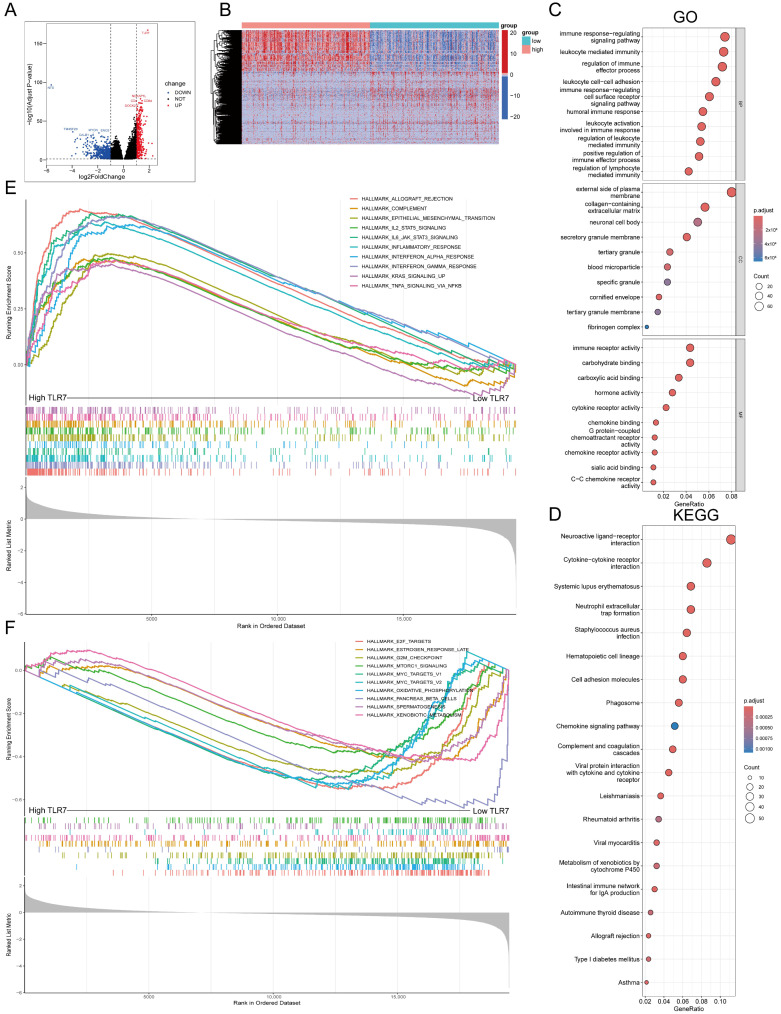
GO, KEGG, and GSEA of high- and low-expressing TLR7 groups in LUAD. (**A**) Volcano plots of DEGs generated by comparing TLR7 groups in LUAD. (|log2FoldChange| > 1 and Adjust *p*-value < 0.05). (**B**) Heatmap of TLR7 high- and low-expression differential genes. The row names of the heatmaps are gene names and the column names are the IDs of samples not shown in the graphs. (**C**,**D**) Differential genes were analyzed for GO and KEGG enrichment, and terms with *p*-adjust < 0.05 were considered significantly enriched. (**E**) The high-expressing TLR7 samples were enriched in the gene set of the HALLMARK collection and only the gene sets with NOM *p* < 0.05 and FDR q < 0.05 were regarded as significant. (**F**) Low TLR7 samples showed enrichment in HALLMARK sets.

**Figure 6 biomedicines-13-00151-f006:**
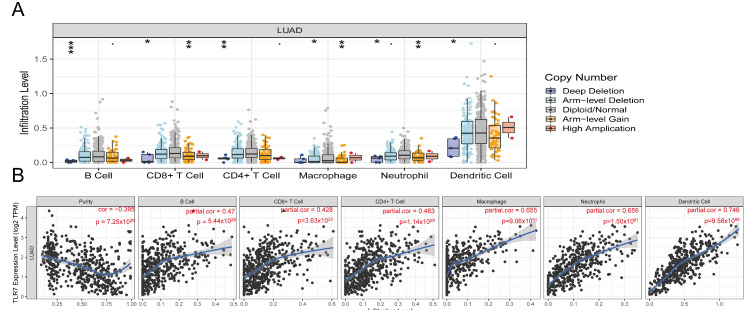
TLR7 expression correlates with infiltrating immune cells in LUAD. (**A**) TLR7 gene copy number variations impact immune cell infiltration. (**B**) TLR7 is linked to lung adenocarcinoma tumor purity and immune cell expression. (**C**) Survival analysis showed that the overall survival of lung adenocarcinoma patients was positively correlated with the expression of B cell, DC, and TLR7. (**D**) Multi-group stacked histogram showing the proportion of 25 tumor-infiltrating immune cells in LUAD tumor samples. (**E**) Bar graph showing the distribution of 28 immune cells between LUAD tumor samples with high or low TLR7 expression. (**F**) Heatmap of the correlation between 28 tumor-infiltrating lymphocytes and TLR7, with the chromaticity of each small colored box representing the corresponding correlation value between the two cells. ***: *p* < 0.001, **: *p* < 0.01, *: *p* < 0.05, ns: not significant.

**Figure 7 biomedicines-13-00151-f007:**
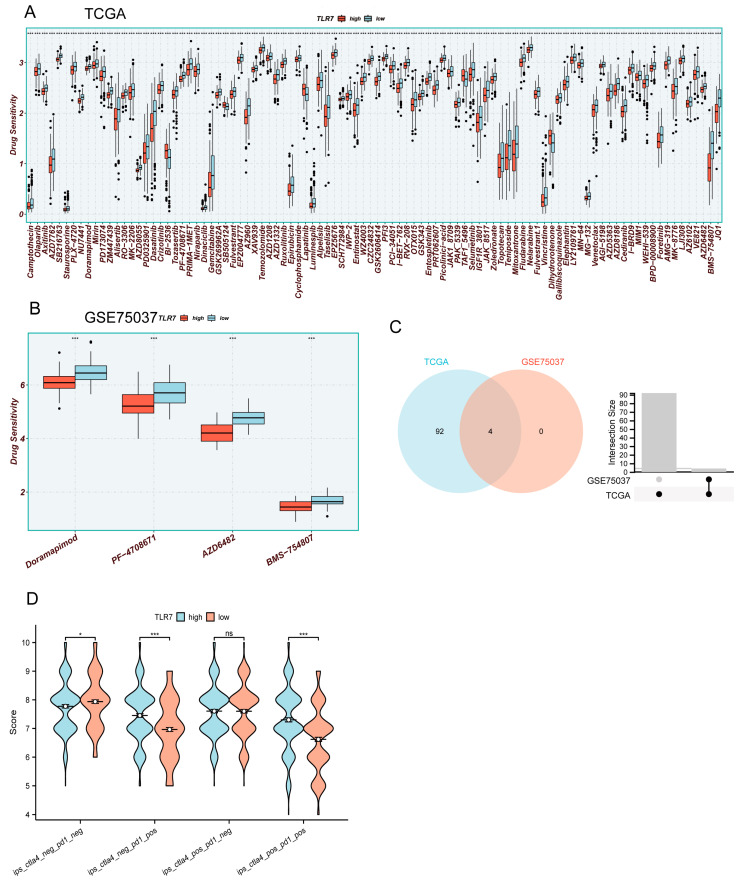
Drug sensitivity and immunotherapy analysis for high and low TLR7 expression. (**A**,**B**) Drug sensitivity analysis of LUAD patients in TCGA and GSE75037. (**C**) Venn analysis of drugs common to (**A**,**B)**. (**D**) IPS analysis of LUAD patients in TCGA. (***: *p* < 0.001, *: *p* < 0.05, ns: not significant).

## Data Availability

The datasets used and/or analyzed during the current study are available from the corresponding author on reasonable request. Data supporting the current findings are available at The Cancer Genome Atlas (TCGA), Gene Expression Omnibus (GEO), University Of California Santa Cruz (UCSC xena), ImmPort, The Cancer Immunome Database (TCIA), The Tumor Immune Estimation Resource (TIMER) and TISIDB websites.

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
