# Peer review of "TLR7: A Key Prognostic Biomarker and Immunotherapeutic Target in Lung Adenocarcinoma"

_biomedicines, 2025, doi:10.3390/biomedicines13010151_

Round 1
Reviewer 1 Report
Comments and Suggestions for Authors
The submitted manuscript entitled: TLR7: A Key Prognostic Biomarker and Immunotherapeutic Target in Lung Adenocarcinoma, resents a robust example of a prototype drug discovery study. High standard of statistics and biostatistics is well noticed. Here we report few points to be addressed before further steps:
Introduction:
-Robust introduction about tumor micro environment.
Materials:
-Line 101: (RNA sequencing data and clinical profiles for 571 LUAD patients). It is not clear how the profiles were selected. What were the inclusion and exclusion criteria?
Results:
-Line 228: More description needed for all steps of figure 1. Also in step 2: is there a specific reason for choosing men in short-pants (Low TLR7 Vs High TLR7)? Please explain
-More description needed for all figure legends
-High standard statistics
-Many confirmational platforms including GEO data, qPCR, WB, and
IHC methods were used.
Discussion
-Well supported and rich discussion
-Line 575: Date or number of the ‘’ethics Committee of the Fourth Military Medical University’’ review should be mentioned.
Author Response
Comments1: Line 101: (RNA sequencing data and clinical profiles for 571 LUAD patients). It is not clear how the profiles were selected. What were the inclusion and exclusion criteria?
Response1: Regarding your question in the review about the selection criteria for RNA sequencing data and clinical information, I provide a detailed answer here. First, our data sources cover two public databases, The Cancer Genome Atlas (TCGA) and Gene Expression Omnibus (GEO), which provide rich genomic, transcriptomic, and clinical data for cancer research. In the TCGA database, we specifically selected RNA sequencing data and clinical information of lung adenocarcinoma (LUAD) patients to ensure that the selected patients had complete clinical and gene expression data to improve the accuracy and reliability of the analysis. In addition, we excluded patients who had already received treatment to eliminate the potential impact of treatment on gene expression data. Our study covered patients with LUAD at (stages I, II, III and IV) in order to comprehensively analyze the progression of the disease at different stages. Similarly, the same was done for the data in the GEO database to ensure consistency and comparability of data. With this approach, we were able to perform an in-depth analysis of gene expression and clinical features of lung adenocarcinoma.
Comments2: Line 228: More description needed for all steps of figure 1.Also in step
Response2: Thank you for your valuable comments. We have carefully considered your suggestion and have described all the steps in Figure 1 in more detail in lines 229-243, marked in red.
Figure 1. Graphical abstract of steps in this study. Step 1: An in-depth analysis of lung adenocar-cinoma samples from the TCGA database using the ESTIMATE algorithm revealed significant correlations between scores on immune and stromal components of the tumor microenvironment and clinical features and patient prognosis, and identified key genes associated with these scores. Step 2: TLR7, identified by PPI and COX analysis, is a key biomarker for LUAD prognosis, and lower tumor expression is associated with poorer prognosis, as validated by GEO. qPCR, WB, and IHC confirmed that TLR7 expression is lower in LUAD compared to normal cells. A Nomogram-based prognostic model incorporating TLR7 can effectively predict patient survival. Step 3: By enrichment analysis, TLR7 was found to be closely associated with immune function. Further immune infiltra-tion analysis showed that TLR7 was positively correlated with the infiltration levels of multiple immune cells. These findings may explain the ability of the tumor microenvironment (TME) to maintain an immunodominant state. Step 4: The response of TLR7 subgroups to immunotherapy was analyzed and patients with high TLR7 were found to be more likely to benefit from anti-PD-1 checkpoint blockade therapy. And four therapeutic candidates for patients with high TLR7 were identified.
Comments3: is there a specific reason for choosing men in short-pants (Low TLR7 Vs High TLR7)?
Response3: Thank you for your question. In this study, the use of the metaphor of “men in shorts” to differentiate between the Low TLR7 and High TLR7 groups has no special meaning. This representation is only to describe the two groups of people with different TLR7 expression levels in order to visualize the results of the study.
Comments4: More description needed for all figure legends
Response4: Thank you for your careful review and valuable suggestions. Based on your feedback, we have added and improved the legends of the charts in the text, and marked in red. The legends of each chart now include more detailed descriptions to ensure that readers can clearly understand the data and analysis results presented in the charts.
Comments5: Line 575: Date or number of the ‘’ethics Committee of the Fourth Military Medical University’’ review should be mentioned.
Response5: Thank you for your careful review and valuable comments. As per your suggestion, we added the specific date or number of the “ethics committee” review on lines 590-593. The section now contains the necessary details to ensure transparency and traceability. Specifically, we have added the following information to the original text: Our research has been approved by the Medical Ethics Committee of the First Affiliated Hospital of the Air Force Medical University. Ethical Committee Name: The Medical Ethics Committee of the First Affiliated Hospital of the Air Force Medical University. Approval Code: KY20213099-1. Ap-proval Date: March 8, 2021. One more thing to clarify with you is that the Air Force Medical University was formerly known as the Fourth Military Medical University, which is the same unit.
Reviewer 2 Report
Comments and Suggestions for Authors In the current comprehensive bioinformatics study, Feiming Hu and colleagues identified an immune-related gene signature associated with LUAD severity, among which TLR7 was found to be one of the most promising marker gene candidates for lung cancer progression. Using a variety of current in silico methods and transcriptomic data, the authors showed that high expression of TLR7 is associated with poor prognosis of LUAD patients and, conversely, with high infiltration of tumor tissue with various immune cells, including both pro-inflammatory and immunosuppressive. Some aspects of the findings were verified experimentally (TLR7 expression in tumor and non-tumor cells and in patient tumor tissue) and using independent GEO datasets. The results obtained by Hu et al. are of great interest in the field of molecular oncology and may be published in the Biomedicines after minor revisions. The main question that should be answered in detail by the authors in the article before publication is why, despite the high representation of immune cells in the TLR7 high group (it should be emphasized - not only immunosuppressive cells, but also pro-inflammatory cells), TLR7 overexpression is still associated with poor survival of LUAD patients? In my opinion, a number of results should be confirmed not only by independent GEO data, but also by using the CCGA database. The following editorial errors should be corrected before publication: line 55 - in my opinion, it would be better to write “and cells that form the lymphatic vessels and the vascular system” since you had cells listed before. line 56 - Add a transcript for CAFs (cancer-associated fibroblasts), as this is the first time CAFs are mentioned in the text of the article. line 107 - please add space line 120 - please add “into” or “as” between grouped and high line 122 - please decipher FC (fold change) line 159 - please decipher FBS (fetal bovine serum) line 165 - please specify which reagent set was used for reverse transcription? lines 174, 205, 279, 411 - insert a space. line 243 and throughout the manuscript - please write numbering of Figures without dots (Figure 2E, but not Figure. 2E) line 264 - please comment briefly on what is included in abnormal data line 272 and throughout the manuscript - in my opinion, ME should be removed from MEturuoise lines 278-279 - the sentence “The results <...> biological processess” is not consistent. Please correct it. line 295 - the phrase “significant genes in 12 genes” sounds incorrect. Please reconstruct the sentence structure. line 324 - move the period after (Figure 3I-M) line 334-338 - difficult to understand sentence structure “As follows, we can <...> ot the patient”. Please write more clearly. line 362 - replace the dot with a space between scores and (Figure 4E). line 376 - replace the dot with a space between patients and (B) line 399 - please decode TILs line 403 - extra dot between cells and (Figure 6A). Please correct. lines 417-418 - incorrect sentence structure “In addition <...> as MDSCs”. Please rewrite more clearly. line 417 - replace the dot with a space between types and (Figure 6F). line 428, Figure 6D - please change memeory with memory Fig. 6F - please enlarge the font as the reader may not be able to see anything. line 450 - (a) please change Venne analysis with Venn diagram analysis; (b) please change "on" with "to identify" (or analogs). lines 459-460 - please decipher the complex names ips_ctla4_neg_pd1_pos and ips_ctla4_pos_pd1_pos. I know what you mean, but the reader of the journal may not understand. Line 480 - please add dot between paclitaxel and Several. Line 500 and throughout the manuscipt - please decide whether you are writing the article in British English (tumour) or American English (tumor). Mixing them in the same text is unacceptable.Author Response
Comments1: The main question that should be answered in detail by the authors in the article before publication is why, despite the high representation of immune cells in the TLR7 high group (it should be emphasized - not only immunosuppressive cells, but also pro-inflammatory cells), TLR7 overexpression is still associated with poor survival of LUAD patients?
Response1: Thank you for your valuable comments and questions. We understand your concern about the relationship between TLR7 overexpression and the prognosis of patients with lung adenocarcinoma (LUAD). In our study, we did observe that TLR7 overexpression was associated with prolonged survival in LUAD patients. Specifically, the results of our data analysis (shown in Figure 3N) support this finding. In Figure 3N, we compared the survival time of LUAD patients with high and low TLR7 expression by survival analysis. The results showed that the group of patients with high TLR7 expression showed longer survival time compared to the low expression group (P = 0.002), suggesting that high TLR7 expression may be associated with a better prognosis. That is, overexpression of TLR7 plays a protective role in LUAD and affects patient survival.
Comments2: In my opinion, a number of results should be confirmed not only by independent GEO data, but also by using the CCGA database.
Response2: Thank you for your suggestion, and we have carefully considered your comments regarding the use of independent data sources to validate the results. In our manuscript, we initially used data from TCGA to screen for differentially expressed genes. Subsequently, we used the TCGA data to establish relationships between TLR7 expression and clinical characteristics as well as survival outcomes. These findings are shown in Figure 3E, F, G, H, and N. In addition, to strengthen the validity of the results, we validated the data from GEO. The results of these analyses are compiled in Supplementary Figure 3. Comparing the results of TCGA and GEO data, we were able to further confirm the correlation between TLR7 expression levels and the prognosis of lung adenocarcinoma patients. We believe that the joint validation of these two databases provides greater reliability and generalizability of our findings. We hope that these additions will fulfill your requirements for validation of results and enhance the persuasiveness of our study.
Comments3: The following editorial errors should be corrected before publication: line 55 - in my opinion, it would be better to write “and cells that form the lymphatic vessels and the vascular system” since you had cells listed before.
Response3: Thank you for your suggestion, at your suggestion, we have changed line 55 in the revised article to read “and cells that form the lymphatic vessels and the vascular system.”
Comments4: line 56 - Add a transcript for CAFs (cancer-associated fibroblasts), as this is the first time CAFs are mentioned in the text of the article.
Response4: Thank you for your suggestion, as per your suggestion, we have added a note in line 57 of the revised article “CAFs (Cancer-associated fibroblasts)”
Comments5: line 107 - please add space line 120 - please add “into” or “as” between grouped and high line 122 - please decipher FC (fold change) line 159 - please decipher FBS (fetal bovine serum) line 165 - please specify which reagent set was used for reverse transcription?
Response5: Thank you for your suggestion, as per your suggestion, we have added spaces at 107 respectively in the revised article, and “into” in line 120 of the revised article, and “log2 FC (fold change)” in line 122 of the revised article,and “FBS (fetal bovine serum)” in line 160 of the revised article. And the reverse transcription reagent we used was HiScript ll Q RT SuperMix for qPCR (+gDNA wiper) (Vazyme Biotech, Nanjing, China), which has been modified in lines 165-166 of the article to read “This RNA was then converted to cDNA by HiScript ll Q RT SuperMix for qPCR (+gDNA wiper) (Vazyme Biotech, Nanjing, China).”
Comments6: lines 174, 205, 279, 411 - insert a space. line 243 and throughout the manuscript - please write numbering of Figures without dots (Figure 2E, but not Figure. 2E)
Response6: Based on your suggestions, we have made changes in the appropriate places in the article and have highlighted them in red.
Comments7: line 264 - please comment briefly on what is included in abnormal data
Response7: Thank you for your question and we would like to provide a brief explanation of the “abnormal data” issue you mentioned. In our study, the term “abnormal data” refers to those data that have a gene expression value of zero.
Comments8: line 272 and throughout the manuscript - in my opinion, ME should be removed from MEturuoise;lines 278-279 - the sentence “The results <...> biological processess” is not consistent. Please correct it. line 295 - the phrase “significant genes in 12 genes” sounds incorrect. Please reconstruct the sentence structure. line 324 - move the period after (Figure 3I-M)
Response8: Thank you very much for your suggestion, we have deleted “MEturuoise” in line 284 of the revised article; in lines 290-291, change to “The results show the top 20 biological processes (BP)”; in line 307, “we identified a significant 12 genes”; and in line 338, we moved the period (Figure 3I-M).
Comments9: line 334-338 - difficult to understand sentence structure “As follows, we can <...> ot the patient”. Please write more clearly.
Response9: Thank you very much for your suggestion, we have revised the revised article at 348-351 to read “As shown in Figure 4A, we incorporated seven indicators to build the prognostic model, and based on the patient's own situation, we can find the corresponding scores, and add these scores to get the total score to assess the corresponding one, three, and five-year survival rate of the patient. -year survival rate of the patient.”
Comments10: line 362 - replace the dot with a space between scores and (Figure 4E). line 376 - replace the dot with a space between patients and (B) line 399 - please decode TILs line 403 - extra dot between cells and (Figure 6A). Please correct.
Response10: Thank you very much for your suggestion, we have revised and marked the revised article in red in the appropriate place. And we have added a note “TILs (Tumor Infiltrating Lymphocytes)” to line 414 of the revised article.
Comments11: lines 417-418 - incorrect sentence structure “In addition <...> as MDSCs”. Please rewrite more clearly.
Response11: I am very sorry, this should have been a sentence that has been changed in lines 433-436 of the revised article “In addition, macrophages, mast cells, memory B cells, monocytes, and various immune cell In addition, macrophages, mast cells, memory B cells, monocytes, and various immune cell subtypes such as natural killer T cells (NKT) and plasmacytoid dendritic cells were also found to be positively correlated with TLR7 expression ( Supplementary Figure 3A).”
Comments12: line 417 - replace the dot with a space between types and (Figure 6F). line 428, Figure 6D - please change memeory with memory Fig. 6F - please enlarge the font as the reader may not be able to see anything. line 450 - (a) please change Venne analysis with Venn diagram analysis; (b) please change "on" with "to identify" (or analogs).
Response12: Thank you very much for your suggestion, we have revised and marked the revised article in red in the appropriate place. And we have also revised line 464 of the revised article to read “Venn diagram” and added “to”.
Comments13: lines 459-460 - please decipher the complex names ips_ctla4_neg_pd1_pos and ips_ctla4_pos_pd1_pos. I know what you mean, but the reader of the journal may not understand.
Response13: Thank you very much for your suggestion, I have annotated lines 474-476 “ips_ctla4_neg_pd1_pos (low level of CTLA-4 expression (neg) and high level of PD-1 expression (pos)) and ips_ctla4_pos_pd1_pos (high level of CTLA-4 expression (neg) and high level of PD-1 expression (pos))”.
Comments14: Line 480 - please add dot between paclitaxel and Several. Line 500 and throughout the manuscipt - please decide whether you are writing the article in British English (tumour) or American English (tumor). Mixing them in the same text is unacceptable.
Response14: Thank you very much for your suggestion, I have added the dot after 495 lines of paclitaxel and have changed all the British English (tumour) to American English (tumor) in the article.
Reviewer 3 Report
Comments and Suggestions for Authors
Dear Authors,
I appreciate the effort and thought put into this study, which explores an important topic in cancer research.
While the manuscript provides valuable insights, there are several areas where improvements could enhance its clarity, rigor, and impact. Below are my key comments and suggestions to improve the article impact to the reader.
Comments:
1. Does the manuscript effectively highlight the knowledge gaps it aims to address regarding lung adenocarcinoma (LUAD) biomarkers?
2. Were the datasets (TCGA, GEO) adequately described, and was the sample size appropriate for drawing statistically significant conclusions?
3. Are the computational methods (ESTIMATE, TIMER, ssGSEA) and statistical analyses (Kaplan-Meier, COX regression) described in enough detail to allow replication as a reproducibility?
4. Does sufficient detail about the experimental validations (qPCR, WB, IHC) for TLR7 expression to assess their reliability?
5. In prognostic Value, Is the evidence convincing that TLR7 expression is significantly correlated with clinical outcomes like survival, TNM stage, and immune cell infiltration?
6. In nomogram Model, Is the prognostic model incorporating TLR7 expression robust and well-validated? Does it provide clinically meaningful predictions?
7. Are the thresholds for differential gene expression and statistical significance (e.g., p-values, FDR) appropriate and justified?
8. Does the identification of four potential therapeutic drugs for high TLR7-expressing LUAD patients provide actionable insights for clinical translation?
9. Does the manuscript adequately address the limitations of the study, such as its reliance on retrospective datasets and limited experimental validation?
10. Are the implications for immunotherapy and personalized medicine clearly discussed, particularly for patients with high TLR7 expression?
11. Were ethical considerations (e.g., use of patient data, informed consent) adequately addressed? Are the claims supported by the stated ethical approvals?
Comments on the Quality of English LanguageThe overall English in the manuscript is clear. The scientific concepts are well-articulated, and technical terms are appropriately used. Sections like methods and results are logically organized and easy to follow.
Author Response
Comments1: Does the manuscript effectively highlight the knowledge gaps it aims to address regarding lung adenocarcinoma (LUAD) biomarkers?
Response1: Thank you very much for your question.In our manuscript, we do address the knowledge gaps in LUAD biomarkers and provide a clear theoretical basis for our research. Specifically, we discuss the limitations of current research in the field and the need for more sensitive and specific biomarkers for early detection and targeted therapy. Our study aims to address these gaps by analyzing survival and the relationship between patients' clinical information and TLR7 expression, finding that early TLR7 expression is higher than late, and that TLR7 has longer survival and more immune cell infiltration. And the establishment of a prognostic model ROC>0.7 can sensitively and accurately predict the survival of patients, all of which provide a basis for TLR7 to become a biomarker for LUAD. We also highlight the potential value of studying TLR7 expression levels in predicting the prognosis of LUAD patients and guiding personalized therapy.
Comments2: Were the datasets (TCGA, GEO) adequately described, and was the sample size appropriate for drawing statistically significant conclusions?
Response2: Thank you for your question. We confirm that the TCGA and GEO datasets we used in this study were adequately characterized and that the sample size was appropriate and sufficient to draw conclusions of statistical significance. First, the TCGA (The Cancer Genome Atlas) and GEO (Gene Expression Omnibus) datasets were used as the primary data sources in our study. In order to ensure the transparency and reproducibility of our study, we have described in detail in our paper the source and content of these two datasets, as well as how we screened and processed these data. The size of the sample size directly affects the statistical significance and reliability of the study results. In our study, we used samples from multiple databases, and we made sure that the sample size was large enough to improve the statistical efficacy of the study and the reliability of the results. We used appropriate statistical methods to analyze the data and multiple databases for mutual validation and ensured that the results obtained were p < 0.05 so that the results were considered statistically significant. In summary, we believe that our study was adequate and appropriate in the description of the data set and the selection of the sample size to support the conclusions of statistical significance that we reached.
Comments3: Are the computational methods (ESTIMATE, TIMER, ssGSEA) and statistical analyses (Kaplan-Meier, COX regression) described in enough detail to allow replication as a reproducibility?
Response3: Thank you for your question. We confirm that the computational methods (ESTIMATE, TIMER, ssGSEA) and statistical analyses (Kaplan-Meier, COX regression) used in this study have been described in detail and are sufficient to support replication of our study by other researchers, ensuring replicability of the study.
Comments4: Does sufficient detail about the experimental validations (qPCR, WB, IHC) for TLR7 expression to assess their reliability?
Response4: Thank you for your question. We confirm that the experimental validation component (including qPCR, WB and IHC), which was described in the original article, ensured the reliability and reproducibility of the TLR7 expression results. The literature we cite describes in detail the specific steps and conditions of these experimental methods, and packages these detailed steps provide a solid experimental basis for our findings and enhance the credibility of our conclusions. We believe that we have adequately demonstrated the reliability of our TLR7 expression results through these detailed experimental validation steps.
Comments5: In prognostic Value, Is the evidence convincing that TLR7 expression is significantly correlated with clinical outcomes like survival, TNM stage, and immune cell infiltration?
Response5: Thank you for your question. To ensure broad applicability and reproducibility of the findings, we analyzed multiple independent databases, including TCGA and GEO. using ESTIMATE and TIMER, we cross-validated the relationship between TLR7 expression and tumor microenvironmental immune cell infiltration in both databases, confirming the relationship between TLR7 and immune response. We built a prognostic model incorporating TLR7 expression and evaluated its predictive ability with an AUC > 0.72, indicating its validity in predicting patient survival. In addition, we validated the expression level of TLR7 in independent samples using qPCR, WB and IHC, which is consistent with the results of our database analysis.
Comments6: In nomogram Model, Is the prognostic model incorporating TLR7 expression robust and well-validated? Does it provide clinically meaningful predictions?
Response6: Thank you for your question. We confirm that the prognostic model incorporating TLR7 expression is robust in nomogram and well validated to provide clinically meaningful predictions. patients with high expression of TLR7 have longer survival and are validated in another dataset (GSE68465). Thus TLR7 is a favorable prognostic factor. Our model was validated by a calibration curve, which yielded a C-index of 0.697 (95% CI: 0.654-0.741) and a pvalue = 4.85e-19. This indicates that the model has a good degree of discrimination. Furthermore, high TLR7 expression levels in melanoma were associated with better clinical outcomes. We also found that patients with high TLR7 gene expression levels were more sensitive to immunotherapy. These findings support the importance and robustness of TLR7 in prognostic modeling.
Comments7: Are the thresholds for differential gene expression and statistical significance (e.g., p-values, FDR) appropriate and justified?
Response7: Thank you for your question. We confirm that the thresholds and statistical significance criteria (p-value, FDR) used for differential gene expression in this study are appropriate and reasonable to ensure the robustness and reliability of our results. Differential gene expression thresholds: we used common screening criteria, i.e., p < 0.05 and multiplicity of change (|log2Fold Chang| > 1). This setting allows for a reasonable determination of statistically significant up- and down-regulated genes, while taking biological significance into account. Statistical significance: we corrected p-values for multiple testing and used False Discovery Rate (FDR) control methods to minimize type I errors. We believe that these choices of methods and thresholds provided a solid statistical basis for the study and ensured the reliability and reproducibility of our results .
Comments8: Does the identification of four potential therapeutic drugs for high TLR7-expressing LUAD patients provide actionable insights for clinical translation?
Response8: Thank you for your question. We confirm that the four potential therapeutic agents identified in this study for patients with high TLR7-expressing lung adenocarcinoma (LUAD) do provide actionable insights that can help in clinical translation. We utilized “oncoppredict” to predict the drug sensitivity of drugs commonly used for LUAD. We obtained 513 LUAD cases from the TCGA database and found that patients with high TLR7 expression were more sensitive to these four drugs (Doramapimod, sf -4708671, AZD6482 and MBS-754807). And we got the same conclusion in another dataset (GSE75037) of 58 LUAD patients. By analyzing a large sample, I believe that our results will have some reliability and provide a basis for clinical translation.
Comments9: Does the manuscript adequately address the limitations of the study, such as its reliance on retrospective datasets and limited experimental validation?
Response9: Thank you for your question. We agree with you when you point out the limitations of our study in relying on retrospective datasets, such as TCGA and GEO, and limited experimental validation.The data provided by TCGA and GEO are retrospective datasets, and these limitations include, among other things, possible inconsistencies in methodology and standards at the time of data collection, the heterogeneity of the patient populations, and the possibility of missing or inaccurately recorded information. To mitigate these limitations, we have taken the following measures: first, we performed rigorous quality control of the data downloaded from TCGA and GEO, including checking for completeness, consistency, and accuracy, as well as removing possible batch effects and outliers to ensure the quality of the data underlying the analyses. Second, to validate the generalizability of the results, we not only used the TCGA dataset, but also combined it with multiple independent cohorts from the GEO dataset (GSE10072, GSE32863, GSE68465, and GSE75037) for validation in order to increase the reliability and reproducibility of the results. This increased the reliability of the conclusions. We also validated TLR7 expression at the RNA level, protein level and immunohistochemical level. We fully understand and recognize the limitations that exist in research. Although we have invested a great deal of effort to minimize the impact of these limitations, we still recognize that no study is perfect. We are confident that these limitations will be overcome over time as the scientific community continues to focus on this area of TLR7 and as more researchers join the field. Our study is one step in this process, and we look forward to future studies that build on ours to further explore and validate the role of TLR7 in LUAD.
Comments10: Are the implications for immunotherapy and personalized medicine clearly discussed, particularly for patients with high TLR7 expression?
Response10: Thank you for your question. Our study revealed an association between TLR7 expression and immune cell infiltration in the tumor immune microenvironment, especially increased immune cell infiltration such as macrophages and CD8 T cells was observed in tumors with high TLR7 expression. This implies that TLR7 may be a promising biomarker for predicting treatment response to ICIs. Using Immunophenoscore (IPS) immune checkpoint data from the TCIA database, we determined the immunogenicity of LUAD and predicted its response to treatment with ICIs. We found that high TLR7 expression was associated with favorable outcomes of ICIs treatment, especially anti-PD-1 therapy, but not CTLA-4 inhibition. This finding supports the potential role of TLR7 in predicting response to immunotherapy and provides a theoretical basis for personalized treatment of LUAD patients with high TLR7 expression.
Comments11: Were ethical considerations (e.g., use of patient data, informed consent) adequately addressed? Are the claims supported by the stated ethical approvals?
Response11: Thank you for your question. We did give due consideration to ethical issues in the study, especially regarding the use of patient data and informed consent. All participating patients signed an informed consent form agreeing to the use of their samples and medical information for research purposes. The use of these data was approved with appropriate ethical approvals. We complied with all applicable ethical guidelines in the study and clearly state this in lines 590-593 of the article. We believe that by doing so, our study adequately considered ethical issues and ensured the legal and compliant use of patient data. We thank you for your interest and believe that our study is ethically defensible.